# A Mixed-Methods Cluster Randomised Waitlist-Controlled Trial of a Goal-Based Behaviour Change Intervention Implemented in Workplaces

**DOI:** 10.3390/ijerph22030398

**Published:** 2025-03-08

**Authors:** Laura Kudrna, James Yates, Lailah Alidu, Karla Hemming, Laura Quinn, Kelly Ann Schmidtke, Janet Jones, Lena Al-Khudairy, Kate Jolly, Paul Bird, Niyah Campbell, Ila Bharatan, Agnieszka Latuszynska, Graeme Currie, Richard Lilford

**Affiliations:** 1Department of Applied Health Sciences, University of Birmingham, Murray Learning Centre, Birmingham B15 2TT, UK; l.alidu@bham.ac.uk (L.A.); k.hemming@bham.ac.uk (K.H.); l.quinn@bham.ac.uk (L.Q.); j.e.jones@bham.ac.uk (J.J.); c.b.jolly@bham.ac.uk (K.J.); paul.bird@uhb.nhs.uk (P.B.); r.j.lilford@bham.ac.uk (R.L.); 2Liverpool Centre for Cardiovascular Science, School of Nursing and Advanced Practice, Liverpool John Moores University, Liverpool L2 2ER, UK; j.yates@2023.ljmu.ac.uk; 3Liberal Arts, College of Arts and Sciences, University of Health Sciences and Pharmacy, St Louis, MO 63110, USA; kelly.schmidtke@uhsp.edu; 4Warwick Medical School, University of Warwick, Medical School Building, Coventry CV4 7AL, UK; lena.al-khudairy@warwick.ac.uk; 5Institute for Mental Health, University of Birmingham, Birmingham B15 2TT, UK; n.campbell@bham.ac.uk; 6Warwick Business School, University of Warwick, Gibbet Hill Road, Coventry CV4 7AL, UK; ila.bharatan@wbs.ac.uk (I.B.); agnieszka.latuszynska@wbs.ac.uk (A.L.); graeme.currie@wbs.ac.uk (G.C.)

**Keywords:** workplaces, occupational health, behaviour change, goal setting, psychology, implementation science

## Abstract

Previous research suggests a goal-based intervention called ‘mental contrasting and implementation intentions’ improves participants’ health and wellbeing. The present study sought to extend these findings to workplaces in the United Kingdom. A mixed-methods cluster randomised controlled trial was conducted with 28 workplaces and 225 staff. All participants deliberated on wishes (potential goals) about improving their health and wellbeing. In the intervention arm, participants were guided to think about the benefits and obstacles to achieving a wish (mental contrasting) and to plan actions to overcome these obstacles (implementation intentions). The results showed no substantive effect of the intervention on average self-reported progress towards what they wished to do for their health and wellbeing four weeks later (mean difference on a 1–7 scale: −0.19; 95% credible interval: −1.08–0.71). Unexpectedly, anxiety increased, and we found evidence that might suggest people identifying as men or of Asian ethnicity made less progress in the intervention group. To explain the results, qualitative focus group data were analysed, guided by normalisation process theory (NPT) and the behaviour change wheel (BCW). Three key themes emerged: insufficient differentiation from other approaches using writing/drawing (NPT), a mismatch between an internal motivational intervention and external barriers (NPT/BCW), and poor timing of opportunities (NPT/BCW). The discussion explores how these results can enhance future workplace health and wellbeing initiatives.

## 1. Introduction

Working and workplaces are important social determinants of health and wellbeing [1]. Staff who are healthy and well are more productive at work (presenteeism), take less time off sick from work (absenteeism), and are less likely to leave their jobs (turnover) [2]. The World Health Organisation estimates 12 billion working days are lost each year to mental ill health alone, and the United Kingdom government reports that ill health costs its economy GBP 150 billion annually [3,4]. Many initiatives aim to improve staff health and wellbeing by modifying working conditions, such as workload and working hours. Others aim to utilise the workplace as a setting for health prevention and promotion, offering support for issues such as mental health, lifestyle health, musculoskeletal health, and chronic diseases. There is substantial scope for these interventions to improve public health and organisational performance; however, the potential for these interventions will be limited to employers and their employees’ willingness to engage. In 2023, Al-Khudairy et al. conducted a cluster randomised controlled trial of an organisational-level monetary incentive to increase uptake of a local government workplace health and wellbeing initiative among employees and employers in the United Kingdom. The results showed that although employees noticed their employers take more action on health and wellbeing in response to the incentive, employees did not change their health behaviours or improve their wellbeing as a result of increased employer action [5]. Therefore, there was a gap between what employers provided and how employees responded.

To improve employee response to the employer provision of initiatives, in this study, we sought to directly target employee motivation. There is a wide psychological literature on improving motivation with goal-based behaviour change interventions [6]. A prominent goal-based intervention demonstrated to be effective across contexts related to health, academic achievement, and pro-social behaviour is ‘mental contrasting plus implementation intentions’ (MCII) [7,8]. In brief, the MCII intervention is a tool that supports meta-cognitive thinking and imagination. Participants identify a health-related ‘wish’ and consider the ideal outcome from achieving the wish. Then, they consider future barriers and make a plan to overcome them (a complete description is provided elsewhere [9]). Our systematic review of MCII found no MCII studies in workplaces outside healthcare [9]. Since conducting the systematic review in December 2021, we identified only one study in the grey literature evaluating MCII techniques in a social care workforce, which included only 24 participants in its analysis [10]. This study had a response rate of less than three percent and did not contain a qualitative component to help understand why the intervention was not effective.

The purpose of this research was to test the effectiveness of using mental contrasting and implementation intentions in workplaces outside the healthcare sector. This addresses the limitation of other research being conducted solely in the health and social care sectors. Our primary objective was to identify effects on subjective perceptions of progress towards achieving goals about health and wellbeing. We also included an embedded qualitative component to identify themes related to the successful implementation of the intervention.

## 2. Materials and Methods

### 2.1. Registration

The trial protocol was published [9] and pre-registered on the ISRCTN (I17828539) and the Open Science Foundation (DOI 10.17605/OSF.IO/8JKVH).

### 2.2. Ethics Statement

Ethical approval was obtained from University of Birmingham’s Science, Technology, Engineering and Mathematics Ethical Review Committee (ERN 21-0744).

### 2.3. Trial Design

We conducted an embedded mixed method [11], waitlisted control, two-arm (“group”), 1:1, cluster randomised controlled trial to evaluate the effects of a goal-based behaviour change intervention on self-reported goal progress, health, and wellbeing. All workplaces had baseline and endline data collection and intervention delivery sessions. Outcome assessors were blind to group allocation during the baseline sessions. Blinding was not possible at endline because the intervention had been received.

### 2.4. Participants

The study was conducted with participants working for organisations based in the West Midlands region of the UK. Their workplaces were enrolled in a publicly provided workplace health and wellbeing initiative (see www.whispas.co.uk, accessed on 19 December 2024). Workplaces were contacted by a university researcher via social media (LinkedIn), email, or over the phone and invited to participate in the study.

### 2.5. Eligibility

Clusters were workplaces located in the West Midlands region of the UK. Workplaces were eligible if they had already signed up to participate in the local government workplace health and wellbeing programme in the West Midlands and if they were willing and able to allow at least three employees to participate in data collection activities. The minimum workplace size was three employees. Employees were eligible if they were aged 16 years or older and willing to provide written consent.

### 2.6. Intervention

The intervention was mental contrasting with implementation intentions and is described in detail elsewhere using the Template for Intervention Description and Replication (TIDieR) checklist [9,12]. Briefly, all staff were encouraged to make a wish to improve their health and wellbeing. In the literature on goal pursuit, a wish is a potential goal that can be implemented through future planning [13]. Therefore, to support turning a wish into a goal, those in the intervention group additionally considered the best future outcome associated with achieving their wish, followed by identifying the main inner obstacle to achieving it (mental contrasting), and, finally, created an ‘if–then’ plan to overcome the obstacle (implementation intentions). Both the intervention and control groups had additional time for questions and group discussion. Exact details of the wording are available in the protocol. The researchers or workplace health and wellbeing leads could deliver the intervention materials, but in all cases, researchers delivered the intervention. Workplaces selected the mode, which was either remote or face-to-face in group sessions.

### 2.7. Control Group

The control group workplaces received ‘usual care’ throughout, meaning that the workplace was able to access information, activities, and support for their health and wellbeing through the local government workplace health initiative. This included activities like health needs assessments, line manager training, and advice on healthy eating.

### 2.8. Outcomes

The primary outcome measure was self-reported progress towards wish (goal) attainment via a seven-point Likert scale to the following item: “So far, how much progress would you say that you have made towards what you wished to do for your health and wellbeing?” (1—no progress to 7—a lot of progress). A related secondary outcome measure was more focussed on behaviour: “And how much progress in changing your behaviour would you say you have made towards what you wished to do for your health and wellbeing?” (1—no progress to 7—a lot of progress). Other secondary measures were self-rated general health, empowerment to change, and psychological wellbeing, including life satisfaction, feelings of happiness and anxiety yesterday, perceived worthwhileness and meaningfulness of activities, optimism, and clear thinking, among other measures [9,14,15,16,17,18,19,20,21,22,23,24].

### 2.9. Sample Size

A sample size of 60 workplaces was expected based on feasibility and a previous cluster randomised trial [5], assuming an exchangeable correlation structure, an average cluster size of 10, varying cluster size (coefficient of variation: 0.30), likely estimates of the intraclass correlation coefficient (ICCs) (0.01, 0.05, and 0.10), and a standard deviation of 1.55 in the progress towards achievement scale [9,25]. Over 80% power is available across all scenarios to detect a half-point increase in the outcome between the intervention and control group. This difference is considered clinically important due to the difficulty in changing health behaviours even when people intend to make a change, see [9].

### 2.10. Randomisation and Blinding

Randomisation was performed by a statistician (LQ) who was not involved in either the recruitment or delivery of the intervention. The randomisation was performed using the random number generator in Stata v16.1. Assignments of workplaces to the intervention or control group were stratified by company size (small, < 50; medium, 51–249; or large, 250+). Sealed envelopes were provided to researchers after baseline data collection and before intervention delivery. All data collectors were blind to group allocation until opening the envelope when delivering the baseline intervention or control session. All participants were recruited before allocation was revealed.

### 2.11. Statistical Methods

The planned quantitative analyses were described in detail elsewhere [9]. In brief, all analyses were intention to treat. Workplace clusters were analysed according to their randomised allocation. Baseline characteristics were summarised using means, standard deviations, medians, interquartile ranges, frequencies, and percentages, as relevant. Bayesian mixed-effects linear regression models were fitted to estimate the intervention effect on self-reported progress towards wish (goal) attainment. The models were fitted using Stata’s bayes prefix command with a sample size of 10,000 MCMC iterations, a burn-in period of 2500, and a single chain. Uninformative vague priors were applied to all model parameters to minimise the influence of prior beliefs: normal (0, 10,000) for fixed effects, normal (0, σ^2^), and inverse gamma (0.001,0.001) for variance components. The model included the intervention indicator (intervention/control), covariates used in the randomisation (company size: small, medium, or large) as fixed effects, and cluster (workplace of participant) as a random effects term. This Bayesian approach allows for the estimation of posterior probabilities for the intervention effect, enabling the calculation of the probability of a clinically important benefit. Posterior mean differences and 95% credible intervals were reported. A fully adjusted covariate model was performed, adjusting for the following covariates: gender, household income, ethnicity, disability, self-rated health, occupational role, home working, health risks, duration in job role, and organisation size. Analysis for secondary outcomes followed the same format; however, posterior probabilities for any benefit (mean difference > 0) and any harm (mean difference < 0) were calculated for outcomes on a different scale from the primary outcome. Additionally, planned subgroup analyses were conducted for the primary outcome according to workplace and demographic characteristics. Subgroup analyses can support intervention delivery and targeting by showing which groups may need additional support with the intervention or do not benefit.

### 2.12. Qualitative Methods

The qualitative methods were described in detail elsewhere [9] and were designed to explain the quantitative findings using an embedded approach [11]. Participants in the intervention groups participated in focus groups to identify themes related to why and how the intervention started to become part of their normal practice or not. Semi-structured topic guides included questions from normalisation process theory (NPT) about purpose of the intervention and influence on health (coherence), improving and sustaining the intervention (cognitive participation), organisational support and incorporation into work and daily life (collective action), and perceptions of the intervention being worthwhile (reflexive monitoring) [26]. NPT was used because it focusses on the evolving approach to adopting new innovations within organisations. Questions about the process of achieving their wish were also included to bring out additional constructs related to individual-level capabilities, opportunities, and motivations identified according to the behaviour change wheel [27], which provides details at individual levels that complement the focus on organisations with NPT. Reflexive thematic analyses were conducted by two authors (LK and JJ), involving familiarisation with the data, coding and generating themes, and refining themes [28]. LK and JJ discussed the data and generated initial codes and themes, mapping these deductively onto the normalisation process theory and behaviour change wheel frameworks [26,27]. LK reviewed these themes and generated the final set of themes, which were subsequently reviewed by KS, JJ, and GC. The results are reported in accordance with the Consolidated Criteria for Reporting Qualitative Research [29].

### 2.13. Protocol Deviations

Several steps were taken to improve recruitment rates and ensure a sufficient sample size for analyses. Originally, only employees in workplaces from Coventry with a minimum of ten staff were eligible to participate. Due to low response rates within Coventry alone, the geographic range was expanded from Coventry to include the broader West Midlands region. The minimum number of employees was also reduced from 10 to 3 because of interest among smaller organisations. Although managers were initially excluded, they wanted to participate too and were placed in separate baseline and endline sessions to employees to reduce social desirability bias. Although 60 workplaces were expected to take part, only 28 expressed interest by the end of the funding period for data collection.

## 3. Results

### 3.1. Participant Flow and Recruitment

Recruitment began in July 2022 and ended in October 2023. There were 28 organisations randomised to the intervention (124 participants) or control group (101 participants) (Figure 1). For the intervention group, 30 participants were lost to follow-up, meaning at endline there were 14 organisations and 94 participants. For the control group, one organisation dropped out after baseline, and 35 participants were lost to follow-up, meaning at endline there were 13 organisations and 66 participants.

### 3.2. Baseline Characteristics

There were 28 organisations at baseline (14 intervention, 14 control) (Table 1). Most participants were females across both groups (75% intervention, 68% control). The average age of participants was slightly higher for the intervention group compared to the control group (41 years versus 36 years, respectively). A higher percentage of participants had a physical or mental health condition in the intervention compared to the control group (37% versus 28%, respectively). Most participants were of White ethnicity in both groups; however, there was a higher percentage of Asian or Asian British participants in the control group (18% versus 6%). Self-rated health was better in the intervention group compared to the control group, with a higher percentage with good health (54% versus 46%).

### 3.3. Primary Outcome—Progress Towards Wish (Goal)

The primary outcome was progress made towards a participant’s wish (goal), measured on a scale from 1 to 7, with higher scores representing more progress. The average progress score was slightly higher in the control group (mean = 4.12, SD = 1.90) compared to the intervention group (mean = 3.87, SD = 2.00) with a mean difference of −0.19 (95% Crl (credible interval): −1.08 to 0.71) (Table 2). The posterior probability for a clinically important benefit (progress towards achieving the stated wish) was only 5.31%. For the fully covariate adjusted model, the mean difference and the posterior probability were very similar.

### 3.4. Subgroup Analysis

Across the different subgroups, there was little evidence of the effect of the intervention on improvement in progress towards participants goals (Appendix A). However, we found evidence that might suggest some potential harm, with harm being defined as instances where the average score was higher in the control group compared to the intervention group. For men, the mean difference was −1.27 (95% Crl: −2.57 to 0.05) with a posterior probability for clinically important harm of 87%. For those of Asian or Asian British ethnicity, the mean difference was −2.24 (95% Crl: −4.50 to 0.02) with a posterior probability for clinically important harm of 93%, and for those with ‘very good’ self-rated health, the mean difference was −1.19 (95% Crl: −2.64 to 0.31) with a posterior probability for clinically important harm of 82%.

### 3.5. Secondary Outcomes

For the outlined secondary outcomes, the effect of the intervention differed (Table 3). For the additional outcome, progress in changing behaviour towards health and wellbeing goals, the average score was higher in the intervention group compared to the control group, with a mean difference of 0.37 (95% Crl −0.42 to 1.18) and a posterior probability of any clinically important benefit of 37%. The intervention increased average feelings of anxiety during the previous day by 1.20 (95% Crl: 0.22 to 2.18), while the posterior probability of any benefit was 1%, meaning that the posterior probability of any harm (increasing anxiety) was 99%.

### 3.6. Qualitative Findings

The completed COREQ (Consolidated Criteria for Reporting Qualitative Research) checklist is given in Appendix B [29]. A summary of the themes is provided in Table 4, which includes some positive perceived benefits. In line with the embedded design, however, we focus below on the three themes that best explained the null or negative findings: competing or similar approaches (aligned with coherence—differentiation, from normalisation process theory—NPT), external contextual resources (aligned with collective action—contextual integration, from NPT; and opportunity, from the behaviour change wheel—BCT), and timing (aligned with enrolment—cognitive participation, NPT, and opportunity, BCW).

#### 3.6.1. Competing or Similar Approaches

Participants described a range of other competing or similar approaches available in and outside the workplace to support them in making progress towards their health and wellbeing goals. These included employee assistance programmes, helplines, and wellness coaching (see Table 4). Other approaches were sometimes described as providing better support for their health and wellbeing than mental contrasting and implementation intentions. For example, one participant described a ‘vision board’ as being more useful because it would enable them to see what they were working towards:

“I think I’d just use a vision board rather than this… I think a vision board for me works better to actually see.”Group 119, medium-sized workplace.

Another participant described the similarity of the intervention to other existing options for working towards their desired health and wellbeing outcomes. When discussing the intervention, they said:

“It’s not dissimilar to what I’d use at work: what’s the outcome I want, what do I need to get that, and what would the barriers be. In that way although it’s not something I’ve used before, it was quite a familiar thing.”Group 117, large-sized workplace.

Although this participant had not used the mental contrasting and implementation intentions approach itself before, it was considered to be similar to what they already used at work and, therefore, not additive to their existing practice. This means that although the intervention may be useful, its approach was not new, and its benefits could not be detected. The lack of differentiation may have diluted the perceived impact. However, this does not explain why some groups had worse outcomes as a result of using the intervention, unless some groups were more aware of other competing innovations.

#### 3.6.2. External Contextual Resources

Another explanation for the null and negative effects found comes from a mismatch between the internal motivational nature of the intervention and employees’ perceived lack of external contextual resources to support their health and wellbeing. Although all employees were in workplaces actively enrolled in the workplace support programme, employees still perceived lack of opportunity to progress on their wish due to factors beyond their control, such as unexpected illness in their wider external housing environment:

“I had a similar goal, it was to start running regularly again. And [then] everyone in my house got sick, and … [laughter] either I was ill or just too tired, or things like that. That’s what got in the way, and I hadn’t anticipated it.”Group 119, medium-sized workplace.

External contexts affected their capability to achieve their wishes and the extent to which the intervention could become integrated as part of their normal practice. Although the instructions of the intervention asked participants to focus on things they could control—inner obstacles that might get in the way of making progress in achieving their wish for their health and wellbeing—in reality, there were external factors that affected whether they could progress towards their goal. A lack of control was described by participants. Two example quotes are provided below.

“Some things can be out of your control, which may be a negative if you think oh no, I haven’t achieved it, but it might have been out of control.”Group 107, small-sized workplace.

“Unexpected obstacles cropped up, which I couldn’t account for and I’ve not been able to. But I know now that is subsiding a little. I know I can put my main wish back into play.”Group 120, medium-sized workplace.

This finding suggests a fundamental mismatch between the motivational underpinning of the intervention and employee needs for support with external factors which could not be anticipated while they were planning. If some groups also experienced more external factors beyond their control, this could explain why the intervention was less effective for them. However, it is also possible that the fundamental attribution error affected participants’ reports, whereby participants rationalised their lack of progress by blaming the situation rather than themselves [30,31]. A stronger motivational intervention may be needed, coupled with external resources for support aligned with employee needs.

#### 3.6.3. Timing

Finally, the timing of the intervention was poor in some cases, which means it was difficult for staff to begin using implementation intentions. A lack of time due to holidays and the competing demands of family and work were reported to limit how much cognitive participation could be undertaken to support initial steps towards using implementation intentions (initiation and enrolment). This is also an opportunity-based factor within the behaviour change wheel, which could be addressed with targeted interventions around timing. For example, one participant described how their wife impeded their ability to progress on their health and wellbeing goals:

“The Mrs works nights [laughter] so I can get more rest and meditate. The obstacle is when she’s got a day off [laughter].”Group 110, small-sized workplace.

Participants were, however, creative about how they could fit in the time by embedding the intervention among other activities:

“At the moment I’m organising a walk to school in May. This [intervention] would fit in quite well, what do I want to achieve, what’s going to be the obstacles and what’s my plan to achieve that at the end of it.”Group 117, large-sized workplace.

“It [the intervention] would be good if we had something that we could plan for in a staff meeting”Group 115, small-sized workplace.

By including the intervention alongside the school run and a staff meeting, participants were ‘bundling’ these behaviours and increasing their likelihood of success. This is similar to temptation bundling, whereby a behaviour that is not enjoyed in the moment (like exercising) is paired with a behaviour that is enjoyed (like listening to an enjoyable podcast) [32]. The timing theme was related to the physical opportunity aspect of the behaviour change wheel (time as a resource) [27] and the enrolment aspect of normalisation process theory (re-organising relationships) [26].

## 4. Conclusions

The results of this research illustrate the importance of rigorously testing workplace health and wellbeing initiatives in randomised trials. Although the benefits of this goal-based behaviour change intervention were well established in other, non-workplace contexts, our largely null results show that the intervention cannot be easily transported into a new workplace context. There are important limits to the generalisation of psychological interventions. There was even some evidence of effects in the opposite direction of what was intended—that is, backfiring [33,34]: overall, feelings of anxiety the previous day increased in the intervention group, and we found evidence that might suggest men and those of Asian ethnicity made less progress when using the intervention. Although some have suggested that the benefits of workplace health and wellbeing interventions are too difficult to demonstrate in randomised trials [35,36], our results show these trials are important because they can identify causal and unanticipated outcomes.

Our complementary investigation of themes aligned with implementation science theories that showed three main plausible explanations for the null and negative findings. First, the implementation intentions intervention was not sufficiently differentiated from other competing approaches like vision boards [37]. Vision boards include drawing to visualise the future rather than only internal meta-cognition and mental visualisations. Future interventions might consider utilising approaches supporting physically self-constructed external representations of goals and progress rather than internal meta-cognition alone [38]. In normalisation process theory, competing similar innovations means that there is a lack of differentiation, which can impede the adoption of the intervention.

Second, although the instructions asked participants to focus on overcoming inner obstacles, participants spoke of the influence of many outer obstacles. Staff reported feeling like they were not in control of achieving their wish, citing factors like unexpected illness. This shows that simply asking is not always enough to modify behaviour. Moreover, it is plausible that the intervention was associated with increased anxiety because participants felt that they were not in control of their outcomes, which is aligned with the prior literature on perceived behavioural control [39].

The finding that participants mentioned outer obstacles is important. A previous trial in the same setting by Al-Khudairy et al. found that increased employer provision of health and wellness initiatives did not result in changes to employees’ health, health behaviour, or wellbeing [5]. In the current trial, there was no direct targeting of external resources to support staff. Instead, staff motivation was directly targeted without any efforts to provide complementary external resources to facilitate progress on achieving goals. Taken together, the results of both trials show that directly targeting employers is seemingly insufficient, as is directly targeting employees. Therefore, we recommend that future efforts should consider aligning employer provision with employees’ health and wellbeing goals, such as through careful health needs assessments that are linked to specific programming.

Finally, participants reported poor timing of the intervention. Prior research has shown that people may be more likely to change their behaviour when their identity or environments change too. For example, people may be more likely to change just before a decade birthday (at 29 or 39, for example), although this evidence is contested [40]. They may be more likely to change during residential relocation too [41]. Bundling the provision of employee health and wellbeing programming with life changes, such as onboarding to a new job, moving offices or homes, or perhaps even before an important birthday for select groups, could be a way to improve the success of efforts when they are implemented.

The results of our study are limited to staff and workplaces who decided to take up the intervention. Despite the issue of selection bias, there may be generalisations of context that make the generalisation of results more likely. The presence of a workplace intervention alone is insufficient as it relies upon employee take-up, as has been shown for the use of flexible leave [42]. Further efforts to encourage the take-up of initiatives are important, as are aligning these efforts with employees’ needs, goals, and preferences, and ensuring good timing of any opportunity. The use of focus groups only is a limitation to our qualitative findings, as participants may have been more likely to disclose personal factors (such as those related to their ethnicity or gender) if individual interviews had been used. These could be important dimensions of experience to better understand in future initiatives and interventions. Using focus groups was largely a pragmatic way to ensure access to workplaces and suggested by the local government partners. Quantitative survey outcomes were self-reported and from a cross-sectional survey. Therefore, reporting heterogeneity could have influenced the results, whereby any effect was diluted because participants had different interpretations of the same scale points [43]. Although we did not achieve our target sample size, limiting our certainty, backfiring effects were still identified, and it is, therefore, appropriate that the trial did not continue further.

Going forward, future research should consider understanding how to better align employer provision with employee needs in workplace health. Local government and other service providers should be cautious about the use of mental contrasting and implementation intentions in workplaces because of the potential for backfiring effects. Instead, efforts should be taken to carefully align the provision of initiatives and services with employees’ interest in improving their health and wellbeing in specific domains. If goal-based behaviour change interventions are provided, they should consider utilising techniques that allow participants to better visualise what they are doing, such as drawing. All provisions should ensure that an initiative is not too similar to existing innovations in other, non-workplace contexts, is delivered in ways that are sensitive to the resources available to staff, and is planned carefully during a time that works well for all.

## Figures and Tables

**Figure 1 ijerph-22-00398-f001:**
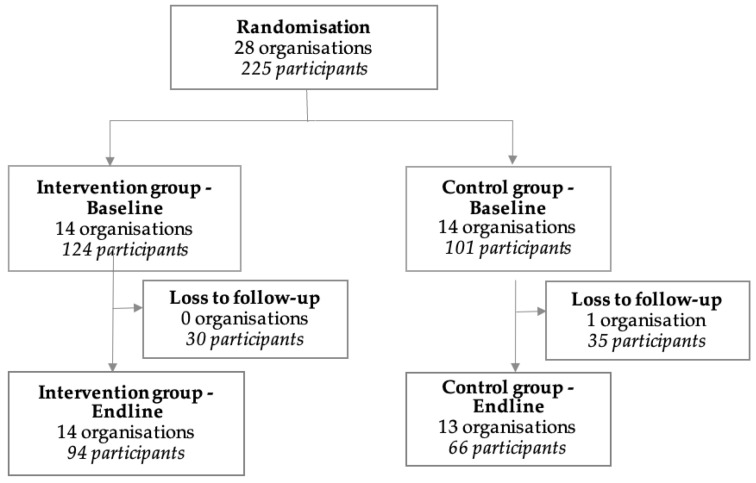
Flow diagram of organisations and participants throughout the trial.

**Table 1 ijerph-22-00398-t001:** Participant characteristics for the intervention and control groups.

Characteristic	Control Group (N = 101)	Intervention Group (N = 124)
**Organisations**	14	14
**Gender, n (%)**		
Female	68 (67.3)	93 (75.0)
Male	30 (29.7)	27 (21.8)
Other	0 (0.0)	2 (1.6)
Missing	3 (3.0)	2 (1.6)
**Age (years), median (IQR)**	36 (28 to 43)	41 (29 to 50)
**Physical or mental health condition lasting 12 months or more**		
Yes	28 (27.7)	46 (37.1)
No	72 (71.3)	77 (62.1)
Missing	1 (1.0)	1 (1.0)
**Ethnicity n (%)**		
White	72 (71.3)	108 (87.1)
Asian or Asian British	18 (17.8)	7 (5.7)
Black, Black British, Caribbean or African	1 (1.0)	4 (3.2)
Mixed or multiple ethnic groups	1 (1.0)	3 (2.4)
Other	5 (5.0)	1 (1.0)
No answer/prefer not to say	3 (3.0)	1 (1.0)
**Self-rated health**		
Very good	16 (15.8)	20 (16.1)
Good	46 (45.5)	67 (54.0)
Fair	37 (36.6)	34 (27.4)
Bad	1 (1.0)	2 (1.6)
Very bad	0 (0.0)	1 (0.8)
Missing	1 (1.0)	0 (0.0)
**Household income**		
<20 k	8 (7.9)	8 (6.5)
20–40 k	35 (34.7)	42 (33.9)
40–60 k	23 (22.8)	34 (27.4)
60–80 k	13 (12.9)	19 (15.3)
80 k+	9 (8.9)	16 (12.9)
Missing	13 (12.9)	5 (4.0)
**Home working**		
Yes	59 (58.4)	89 (71.8)
No	40 (39.6)	33 (26.6)
Missing	2 (2.0)	2 (1.6)
**Feeling physically safe at work**		
Strongly agree	55 (54.5)	74 (59.7)
Agree	34 (33.7)	46 (37.1)
Neither agree nor disagree	8 (7.9)	4 (3.2)
Disagree	3 (3.0)	0 (0.0)
Strongly disagree	1 (1.0)	0 (0.0)
**Duration in job role**		
<2 yrs.	46 (45.5)	51 (41.1)
2–5 yrs.	23 (22.8)	27 (21.8)
5–10 yrs.	20 (19.8)	27 (21.8)
>10 yrs.	11 (10.9)	19 (15.3)
Missing	1 (1.0)	0 (0.0)
**Organisation size**		
Small	56 (55.5)	48 (38.7)
Medium	21 (20.8)	59 (47.6)
Large	24 (23.8)	17 (13.7)
**Occupational role**		
Professional	38 (37.6)	46 (37.1)
Non-professional	60 (59.4)	78 (62.9)
Missing	2 (3.0)	0 (0.0)
**Manager**		
Yes	15 (14.9)	31 (25.0)
No	83 (82.2)	93 (75.0)
Missing	3 (3.0)	0 (0.0)
**Company sector**		
Manufacturing, commercial, or manual	14 (13.9)	9 (7.3)
Services	37 (36.6)	47 (37.9)
Social, public, or intellectual	50 (49.5)	68 (54.8)

**Table 2 ijerph-22-00398-t002:** Mean difference between the control and intervention group on progress towards wishes (goals) for primary outcome.

	Control Group	Intervention Group	Mean difference (95% Credible Interval)	Posterior Probability of Benefit (Mean Difference > 0.5)	Posterior Probability of Harm (Mean Difference < 0.5)
	Progress Towards Wish (Scale 1–7) Mean (SD)				
**Unadjusted model ***	4.12 (1.90)	3.87 (2.00)	−0.19 (−1.08 to 0.71)	5.31%	21.73%
**Covariate adjusted model ****	4.22 (1.90)	3.90 (2.03)	−0.19 (−1.06 to 0.73)	6.56%	24.70%

Note: Mean differences and 95% credible intervals were reported along with the posterior probability of clinically important benefit (mean difference > 0.5) and clinically important harm (mean difference < 0.5). * Estimates not given by model due to low number of observations in categories. ** Model adjusted for the following covariates: gender, household income, ethnicity, disability, self-rated health, occupational role, home working, health risks, duration in job role, and organisation size.

**Table 3 ijerph-22-00398-t003:** Mean difference between the control and intervention group on progress towards wishes (goals) for secondary outcomes.

	Control Group	Intervention Group	Mean Difference (95% Credible Interval)	Posterior Probability (Mean Difference > 0.5)
	Mean (SD)			
**Progress in changing behaviour towards health and wellbeing goal (1–7)**	3.58 (1.78)	3.65 (1.89)	0.37 (−0.42 to 1.18)	36.92%
**Self-rated health (1–5)**	3.98 (0.77)	3.82 (0.77)	−0.16 (−0.38 to 0.07)	8.38%
**Perceptions of health and wellbeing (0–10)**				
Subjective wellbeing	7.13 (1.37)	7.20 (1.72)	0.05 (−0.50 to 0.65)	56.88%
Satisfied with job	7.02 (2.10)	7.33 (1.74)	0.18 (−0.33 to 0.74)	75.02%
Happy yesterday	7.06 (1.96)	6.76 (2.19)	−0.22 (−0.93 to 0.52)	27.17%
Anxious yesterday *	3.49 (2.83)	4.69 (2.85)	1.2 (0.22 to 2.18)	0.98%
Work activities meaningful	7.38 (2.21)	7.35 (1.84)	0.06 (−0.51 to 0.64)	58.55%
Work activities enjoyable	7.09 (1.95)	7.11 (1.86)	0.08 (−0.47 to 0.64)	60.58%
**Mental wellbeing, statements about feelings and thoughts (1–5)**				
Feeling optimistic about future	3.53 (0.95)	3.58 (0.74)	0.02 (−0.32 to 0.39)	55.65%
Feeling relaxed	3.10 (0.82)	3.11 (0.77)	0.12 (−0.17 to 0.42)	80.12%
Dealing with problems well	3.55 (0.75)	3.50 (0.74)	0.08 (−0.19 to 0.35)	70.85%
Thinking clearly	3.45 (0.72)	3.54 (0.76)	0.29 (0.01 to 0.59)	97.74%
Feeling close to other people	3.70 (0.91)	3.66 (0.78)	0.02 (−0.28 to 0.33)	53.48%
Able to make up own mind about things	3.79 (0.88)	3.88 (0.78)	0.13 (−0.21 to 0.50)	78.62%
**Agreement with following statements about health and wellbeing (1–5)**				
Confident about ability to look after health and wellbeing	3.91 (0.97)	3.84 (0.78)	0.03 (−0.27 to 0.36)	56.70%
Know what to do to improve health and wellbeing	4.17 (0.87)	4.01 (0.69)	0.01 (−0.27 to 0.30)	52.51%
Handle problems well when they arise	3.75 (0.94)	3.57 (0.88)	−0.07 (−0.40 to 0.30)	34.12%
Confident to make the best choices to look after health and wellbeing	3.83 (0.94)	3.73 (0.80)	0.06 (−0.28 to 0.43)	61.87%
Feel physically safe at work	4.32 (0.87)	4.58 (0.57)	0.1 (−0.15 to 0.36)	79.00%

* Higher scores indicate worse wellbeing.

**Table 4 ijerph-22-00398-t004:** Qualitative themes from normalisation process theory.

Coherence (Sense-Making) ^1^	*Positive*	*Negative*
**Differentiation**	Unique from employee assistance programmes (EAPs), health cash plan providers, helplines, mental health training, and other coaching models. Consistent with wellbeing focus on employee appraisals.	Not distinct from vision boards and similar approaches already implicitly used at work and in personal life.
Individual specification	WOOP ^2^ seen as being about prioritising wellbeing, relatively simple and feasible, use of writing for steps, follow-through, and proactiveness.	Hard to relate idea of objectives to personal life.
Internalisation	Value in breaking down steps, stating obstacles helpful, focussing on one goal in a short timeframe, goal formation and discipline, prioritises own feelings/thoughts/needs, and provides perspective.	‘Wish’ not perceived as important versus ‘goal’, but avoiding the word ‘goal’ is helpful as ‘wish’ is gentler and the word ‘WOOP’ ^2^ is a celebration.
**Cognitive participation (engagement)**		
Initiation	Writing down initial steps, staying focussed.	None identified.
Enrolment	Discussing progress with colleagues, doing a little at a time, redirecting attention, cutting out unhelpful thoughts, and removing excuses.	Timing poor—over holidays not a good time.
Legitimisation	Provided space to focus on wish, helps support self over others.	Need to support others versus self.
Activation	Overcome procrastination with WOOP ^2^ steps, pushing aside obstacles.	Unclear if WOOP ^2^ is beneficial in the long term.
**Collective action (work to enable intervention to happen)**		
Interactional workability	Easier when family away or break from work.	Hard to prioritise when competing with demands of family and work.
Relational integration	More effort as group, encouragement from others, and supportive team.	Did not discuss WOOP ^2^ with colleagues.
Skill set workability	No assistance from work was required or provided.	None identified.
Contextual integration	None identified.	More external resources needed to achieve goals; working from home leads to forgetting WOOP ^2^.
**Reflexive monitoring (formal and informal appraisal of benefits and costs of intervention) ^3^**		
Communal appraisal	Team created WOOP^2^ check-ins every four weeks (after intervention).	None identified.
Individual appraisal	Value in building confidence, enablement, having a good why, breaking wish down and identifying obstacles, short-term approach, simplicity, and further applying to working life.	Wish too hard to measure, WOOP ^2^ too easy/too much work, success depends on type of wish, more motivation and time needed, and more flexibility for multiple wishes needed.
Reconfiguration	Has use as coaching and mentoring tool, used again when first wish did not work.	Add grounding/relaxation techniques, journalling, reminder system (phone, paper, check-ins, and reviews), incentives/consequences, likeminded group of people as support, memorable piece of paper or visualisation, professional coach, more line manager support, and make it longer.

^1^ Communal specification (under coherence), no themes identified; ^2^ WOOP stands for ‘wish’, ‘outcome’, ‘obstacle’, ‘plan’ and is another name for the intervention (‘mental contrasting plus implementation intentions’); ^3^ Systematisation (under reflective monitoring), no themes identified.

## Data Availability

The anonymous quantitative data are available upon request from Laura Kudrna, l.kudrna@bham.ac.uk.

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
