# Peer review of "A Mixed-Methods Cluster Randomised Waitlist-Controlled Trial of a Goal-Based Behaviour Change Intervention Implemented in Workplaces"

_ijerph, 2025, doi:10.3390/ijerph22030398_

Round 1

Reviewer 1 Report

Comments and Suggestions for Authors

Please make corrections according to the attached document.

Author Response

Comment 1:  Introduction and rationale. Enhance the Contextual Background: Provide more detailed insights into why workplace interventions are critical, linking this to broader public health and organizational performance contexts.

Reply 1: Thank you for this comment. We have now highlighted that: “There is substantial scope for these interventions to improve public health and organisational performance; however, the potential for these interventions will be limited to employers and their employees’ willingness to engage.”

Comment 2:   Clarify Gaps in Literature: Explicitly highlight how this study addresses a significant gap, particularly in applying goal-based interventions in non-healthcare workplaces.

Reply 2: We now clarify that “This addresses the limitation of other research being conducted solely in the health and social care sectors.”

Comment 3: Strengthen Sample justification: The reduced number of workplaces (28 instead of the expected 60) may impact the generalizability of results. Discuss strategies for mitigating this limitation and how it affects statistical power.

Reply 3: This is an important point. We now clarify in the discussion that: “Although we did not achieve our target sample size, limiting our certainty, backfiring effects were still identified, and it is, therefore, appropriate that the trial did not continue further.” We have now also clarified that we modified participant eligibility criteria to mitigate the limitation of low sample size (see Comment 4 reply).

Comment 4:  Address Inclusion Criteria: Provide more detailed reasoning for the modifications to participant eligibility, such as including smaller organizations and managers, to ensure readers understand the rationale.

Reply 4: We now clarify that “Several steps were taken to improve recruitment rates and ensure a sufficient sample size for analyses.”

Comment 5: Intervention Details. Provide practical Examples: Include more concrete examples or scenarios illustrating how the intervention was implemented in workplaces, especially the "mental contrasting and implementation intentions" approach.

Reply 5: It is important that this information is available, and the full details are provided in the published protocol supporting materials. Please see here: https://journals.plos.org/plosone/article/file?type=supplementary&id=10.1371/journal.pone.0282848.s003. We now note that “Exact details of the wording are available in the protocol.”

Comment 6: Control Group Engagement: Clarify what activities or resources were available to the control group during the intervention period to strengthen the comparison.

Reply 6: Thank you for this suggestion. We have now added a section on the control group: “The control group workplaces received ‘usual care’ throughout, meaning that the workplace was able to access information, activities, and support for their health and wellbeing through the local government workplace health initiative. This included activities like health needs assessments, line manager training, and advice on healthy eating.”

Comment 7: Detail Subgroup Analysis: Expand on the reasoning behind the chosen subgroups (e.g., gender, ethnicity) and discuss implications for differential outcomes across these groups.

Reply 7: We now clarify that “Subgroup analyses can support intervention delivery and targeting by showing which groups may need additional support with the intervention or do not benefit.” Gender and ethnicity are common subgroups considered for interventions that stand to impact public policy.

Comment 8:  Highlight statistical Methods: Provide additional clarity on the Bayesian mixed-effects regression models used, especially for readers less familiar with these approaches.

Reply 8: We have added the following clarifying details: “Bayesian mixed effects linear regression models were fitted to estimate the intervention effect on self-reported progress towards wish (goal) attainment. The models were fitted using Stata's bayes prefix command with a sample size of 10,000 MCMC iterations, a burn-in period of 2,500, and a single chain. Uninformative vague priors were applied to all model parameters to minimise the influence of prior beliefs: normal (0, 10,000) for fixed effects, normal (0, σ^2), and inverse gamma (0.001,0.001) for variance components. The model included the intervention indicator (intervention/control), and covariates used in the randomisation (company size: small, medium, large) as fixed effects, and cluster (workplace of participant) as a random effects term. This Bayesian approach allows for the estimation of posterior probabilities for the intervention effect, enabling the calculation of the probability of a clinically important benefit. Posterior mean differences and 95% credible intervals were reported.”

Comment 9: Results interpretation • Discuss Null Findings: While the null results are significant, discuss whether the lack of differentiation from other workplace interventions diluted the perceived impact.

Reply 9: We now state that “The lack of differentiation may have diluted the perceived impact.”

Comment 10:  Explore negative Outcomes: Provide more depth on why anxiety levels increased in the intervention group and how this aligns with or diverges from previous studies.

Reply 10: We now hypothesise that “Moreover, it is plausible that the intervention was associated with increased anxiety because participants felt that they were not in control of their outcomes, which is aligned with prior literature on perceived behavioural control.”

Gallagher, M. W., Bentley, K. H., & Barlow, D. H. (2014). Perceived control and vulnerability to anxiety disorders: A meta-analytic review. Cognitive Therapy and Research, 38, 571-584.

Comment 11: Qualitative Insights • Link Themes to practice: Translate key qualitative themes (e.g., external barriers, timing issues) into actionable recommendations for designing future interventions.

Reply 11:  We make this translation in the Conclusion and have now expanded it based on your comment, stating that “If goal-based behaviour change interventions are provided, they should consider utilising techniques that allow participants to better visualise what they are doing, such as drawing. All provision should ensure that an initiative is not too similar to existing innovations in other, non-workplace contexts, be delivered in ways that are sensitive to the resources available to staff, and are planned carefully during a time that works well for all.” If the Editor prefers us to move this text, we are open to moving the text to the Results.

Comment 12: Address Diversity of Experiences: Reflect on how demographic factors (e.g., ethnicity, workplace size) influenced participants' experiences and outcomes.

Reply 12: Unfortunately, we did not ask how demographic factors were linked to participants’ experiences and outcomes in the qualitative research. This was because of the focus group approach and is a limitation, which we now note in the Discussion: “The use of focus groups only is a limitation to our qualitative findings, as participants may have been more likely to disclose personal factors (such as those related to their ethnicity or gender) if individual interviews had been used.”

Comment 13:   Conclusion and implications • Broaden recommendations: Consider offering strategies for aligning workplace interventions with organizational culture and employee preferences, emphasizing timing and resource availability.

Reply 13: In the Conclusion, we are now more specific on offering strategies and state, “If goal-based behaviour change interventions are provided, they should consider utilising techniques that allow participants to better visualise what they are doing, such as drawing.” We also note that: “All provision should ensure that an initiative is not too similar to existing innovations in other, non-workplace contexts, be delivered in ways that are sensitive to the resources available to staff, and are planned carefully during a time that works well for all.” We believe this is as broad as the data will allow us to go without over-interpreting, however, we are open to further modifying this if the Editor recommends.

Comment 14: Policy implications: Highlight potential implications for workplace health policies, especially for organizations in the West Midlands.

Reply 14: This is a good point, and we now state that “Going forward, local government and other service providers should be cautious about the use of mental contrasting and implementation intentions in workplaces because of the potential for backfiring effects.” Because local government runs similar programmes throughout the UK, we decided not to focus on the West Midlands specifically (please see whispas.co.uk).

Comment 15:  Limitations and Future Research • Transparency in limitations: Acknowledge the limitations of the focus group methodology and the potential bias introduced by participant selection.

Reply 15: About the focus groups, we state:

“The use of focus groups only is a limitation to our qualitative findings, as participants may have been more likely to disclose personal factors (such as those related to their ethnicity or gender) if individual interviews had been used. These could be important dimensions of experience to better understand in future initiatives and interventions.”

About the participant selection, we now state:

“The results of our study are limited to staff and workplaces who decided to take up the intervention. Despite these issues of selection bias, there may be generalisations of context that make the generalisation of results more likely.”

Comment 16: Suggest Future directions: Propose specific areas for future research, such as tailoring interventions to different workplace cultures or integrating external resources to address barriers.

Reply 16: Thank you for this comment. We have expanded our final paragraph to include a comment on future research directions: “Going forward, future research should consider understanding how to better align employer provision with employee need in workplace health.”

Reviewer 2 Report

Comments and Suggestions for Authors

The subject of the manuscript is relevant. However, it has major limitations that need to be improved. Specifically, they are the following:

In the Introduction section, the subject of study needs to be presented in greater depth based on previous literature. 

On the other hand, the material and method is adequate, presenting the results in a clear and orderly manner. However, although the conclusions are presented based on the findings of the present study, the discussion section has not been included. In this sense, its inclusion would be relevant, discussing the results found with other lines of research. 

Therefore, the following two modifications are to be made:

1. to present the introduction with a greater depth of literature on the topic of study.

2. Include a discussion section, discussing the results found with other lines of research.

Author Response

Comment 1. The subject of the manuscript is relevant. However, it has major limitations that need to be improved. Specifically, they are the following: In the Introduction section, the subject of study needs to be presented in greater depth based on previous literature. 

Reply 1. Thank you for this comment. Based on a comment from Reviewer 3, who commented positively on the introduction, we have added an additional reference that reviews motivation and behaviour change interventions. Based on a comment from Reviewer 1, we have also more clearly articulated the gap in prior literature. However, we are open to expanding our Introduction further if the Editor feels it is important to do so.

Comment 2: On the other hand, the material and method is adequate, presenting the results in a clear and orderly manner. However, although the conclusions are presented based on the findings of the present study, the discussion section has not been included. In this sense, its inclusion would be relevant, discussing the results found with other lines of research. Therefore, the following two modifications are to be made: 1. to present the introduction with a greater depth of literature on the topic of study. 2. Include a discussion section, discussing the results found with other lines of research.

Reply 2: Thank you for this comment. Apologies for any confusion. We have followed the format of the journal included our ‘Discussion’ material in a section titled ‘Conclusion’ . We include over 10 references to other research in the Conclusion, however, we are open to adding more of the Editor feels this is necessary.

Reviewer 3 Report

Comments and Suggestions for Authors

Abstract

- Line 17: "helps people improve their health and wellbeing". Does this statement refer to all people, or a specific population? I think it is a big statement to make if the statement implies all people.

- Line 20: is making a with the same as establishing a goal? If so, I think this needs to be clarified or more effectively articulated. In relation to "making a wish", this is something that people do often - I would argue that establishing a goal is different to making a wish.

A good quality Introduction - well done.

- Lines 57-58: If there is a wide psychological literature on improving motivation with goal-based change interventions then I believe references are needed.

Materials and Methods

There is some information in section 2.12 about organisations and participants, but there is little information about how participants were recruited. How were potential participants made aware of this opportunity, how was this communicated, and how did potential participants express interest?

- Intervention (2.6): I am uncertain about the notion or concept of a wish. I would argue that a wish may be viewed by some as aspirational or a fantasy, and a short-term/acute focus that has little responsibility or accountability inherent within it. Is a wish framed upon any form of theoretical framework? In light of this, the notion or concept of a goal and setting goal incorporates structure and is framed upon a theoretical foundation.

The lack of structure/framework to guide participants in relation to a wish leads me to think that individual responsibility, accountability, and adherence may be at risk (or questionable), and brings to question authenticity.

2.7 (Outcomes)

Were participants provided any guidance or education regarding the self-report likert scale options? A lack of participant knowledge and understanding can lead to inaccurate reporting (under or over reporting). 

2.11 Qualitative Methods

 - Line 167: I have concerns about an intervention becoming part of normal practice considering that the intervention period was only four weeks. There is a healthy amount of literature in the field of habits and behaviour change, but with such a short time frame this does not allow for adequate time for such a measurement.

 - Line 178: why were the normalisation process theory and behaviour change wheel frameworks used? More details and justification needed here.

Results

3.4 Subgroup analysis

- What is meant by potential harm?

3.5 Secondary outcomes

- It would be beneficial to have more information about anxiety and what this related to in the study. Anxiety in relation to what exactly?

3.6.1

- Lines 279-284: how do you know that the principles of mental contrasting were considered to be normal and not additive to participants' existing practice? How was this measured/gauged?

- Why could benefits (or perceived benefits) not be detected? This is a major implication and warrants explanation, clarification, and direction.

Conclusions

- Lines 372-373: "although the intervention asked participants to focus on overcoming inner obstacles, participants spoke of the influence of many outer obstacles".

How did the intervention ask this? I am not sure that an intervention can ask this, but stipulations or regulations around how an intervention is designed and implemented can imply this. I think this needs explanation and clarification. 

Moreover, this demonstrates that simply asking is not enough to modify behaviour or to expect adherence.

Author Response

Comment 1: Abstract, Line 17: "helps people improve their health and wellbeing". Does this statement refer to all people, or a specific population? I think it is a big statement to make if the statement implies all people.

Response 1: Thank you for the suggestion to be more precise. We agree, and have included the word ‘participants’ to clarify it is only participants in research studies.

Comment 2: Abstract, Line 20: is making a with the same as establishing a goal? If so, I think this needs to be clarified or more effectively articulated. In relation to "making a wish", this is something that people do often - I would argue that establishing a goal is different to making a wish.

Response 2: Thank you for drawing the distinction between wish and goal, which is something that we considered at various stages of the resesarch. We agree that making a goal is different to making a wish and draw on text from Gollwitzer and Oettingen (2011), below. At our protocol review stage, we reflected based on reviewers’ comments with our steering committee and collaboratively decided to use “goal-based” in the title, as wishes are potential goals. However, the WOOP intervention uses the language wish, and, therefore, we use this word throughout the manuscript. We have now clarified in the abstract of this manuscript that wishes are potential goals, as per the existing literature quoted below:

“The concept of implementation intentions grew out of a more comprehensive approach to goal pursuit: the mindset theory of action phases (Gollwitzer, 1990). The mindset model of action phases sees successful goal pursuit as solving a series of successive tasks: deliberating on wishes (potential goals) and choosing between them; planning and initiating goal-directed actions; bringing goal pursuit to a successful end; and evaluating its outcome. This task notion implies that people can activate cognitive procedures (mindsets) that facilitate task completion simply by getting heavily involved with the task at hand. Whereas deliberating between potential goals (i.e., wishes) activates cognitive procedures (i.e., a deliberative mindset) that facilitate decision making, engaging in planning activates those procedures (i.e., an implemental mindset) that support the implementation of goals (p. 177).”

Gollwitzer, P. & Oettingen, G. (2011). Planning promotes goal striving. In K. Vohs & R. Baumeister (Eds.), Handbook of self-regulation: research, theory, and applications (pp. 162-185). ISBN 978-1-606-23948-3

Our protocol reviews, which address this issue, are available here: https://journals.plos.org/plosone/article/peerReview?id=10.1371/journal.pone.0282848

Comment 3: A good quality Introduction - well done. Lines 57-58: If there is a wide psychological literature on improving motivation with goal-based change interventions then I believe references are needed.

Response 3: Thank you. We agree a reference is needed and have added a review that includes studies on improving motivation with goal-based behaviour change interventions:

Gillison, F. B., Rouse, P., Standage, M., Sebire, S. J., & Ryan, R. M. (2019). A meta-analysis of techniques to promote motivation for health behaviour change from a self-determination theory perspective. Health Psychology Review, 13(1), 110-130.

Comment 4: Materials and Methods. There is some information in section 2.12 about organisations and participants, but there is little information about how participants were recruited. How were potential participants made aware of this opportunity, how was this communicated, and how did potential participants express interest?

Reply 4: In all cases, workplaces were contacted by a University researcher via social media (LinkedIn), email, or over the phone and invited to participate in the study. This detail has been added to section 2.4: “Workplaces were contacted by a University researcher via social media (LinkedIn), email, or over the phone and invited to participate in the study.”

Comment 5: Intervention (2.6): I am uncertain about the notion or concept of a wish. I would argue that a wish may be viewed by some as aspirational or a fantasy, and a short-term/acute focus that has little responsibility or accountability inherent within it. Is a wish framed upon any form of theoretical framework? In light of this, the notion or concept of a goal and setting goal incorporates structure and is framed upon a theoretical foundation. The lack of structure/framework to guide participants in relation to a wish leads me to think that individual responsibility, accountability, and adherence may be at risk (or questionable), and brings to question authenticity.

Reply 5: Thank you for raising the important issue of the definition of a wish and its theoretical framework. Yes, a wish is based on a theoretical framework about goal pursuit. This draws on the work of Gollwitzer and Oettigen (2011), as above, who developed the intervention. We now clarify that wishes are potential goals in section 2.6 and note that this comes from existing literature. The language around ‘wish’ versus ‘goal’ was a point of much discussion throughout the research and we appreciate the opportunity to clarify the meaning and theoretical underpinning of wish in the academic paper.

Comment 5: 2.7 (Outcomes) Were participants provided any guidance or education regarding the self-report likert scale options? A lack of participant knowledge and understanding can lead to inaccurate reporting (under or over reporting). 

Reply 5: This is a good point. Other than labelling the scale points, education was not provided. We now note the limitations of the self-report scale questions: “Quantitative survey outcomes were self-reported and from a cross-sectional survey. Therefore, reporting heterogeneity could have influenced the results, whereby an effect was diluted because participants had different interpretations of the same scale points.”

Rice, N., Robone, S., & Smith, P. (2011). Analysis of the validity of the vignette approach to correct for heterogeneity in reporting health system responsiveness. The European Journal of Health Economics, 12, 141-162.

Comment 6: 2.11 Qualitative Method Line 167: I have concerns about an intervention becoming part of normal practice considering that the intervention period was only four weeks. There is a healthy amount of literature in the field of habits and behaviour change, but with such a short time frame this does not allow for adequate time for such a measurement.

Reply 6: This is a good point, and we now stated “started to become” rather than “became” normal practice.

Comment 7: Line 178: why were the normalisation process theory and behaviour change wheel frameworks used? More details and justification needed here.

Reply 7: We have now added this detail: “NPT was used because it focusses on the evolving approach to adopting new innovations within organisations” and “[the behaviour change wheel] provides details at individual levels that complement the focus on organisations with NPT.”

Comment 8: 3.4 Subgroup analysis. - What is meant by potential harm?

Reply 8: We have now defined harm as “as instances where the average score was higher in the control group compared to the intervention group.”

Comment 9: 3.5 Secondary outcome.  It would be beneficial to have more information about anxiety and what this related to in the study. Anxiety in relation to what exactly?

Reply 9: Anxiety is classified as a general mood-state that is not object directed, i.e., anxiety is not a reaction to a particular thing. The exact item is, “Overall, how anxious did you feel yesterday?” (0-10), which is contained in our protocol. On line 130, this is defined as “anxiety yesterday”, and we now clarify in the secondary outcome section that the measure is “average feelings of anxiety yesterday.”

Comment 10: Lines 279-284: how do you know that the principles of mental contrasting were considered to be normal and not additive to participants' existing practice? How was this measured/gauged?

Reply 10: This is based on the participant statement that “It’s not dissimilar to what I’d use at work.” We have revised the text to more closely align with the quote: “Although this participant had not used the mental contrasting and implementation intentions approach itself before, it was considered to be similar to what they already used at work, and, therefore, not additive to their existing practice.”

Comment 11: Why could benefits (or perceived benefits) not be detected? This is a major implication and warrants explanation, clarification, and direction.

Reply 12: Apologies for the lack of clarity. There were perceived benefits and these are listed in column 2 of Table 4. We now clarify this in 3.6: “A summary of the themes is in Table 4, which includes some positive perceived benefits. In line with the embedded design, however, we focus below on the three themes that best explained the null or negative findings”

Comment 12: Conclusions Lines 372-373: "although the intervention asked participants to focus on overcoming inner obstacles, participants spoke of the influence of many outer obstacles". How did the intervention ask this? I am not sure that an intervention can ask this, but stipulations or regulations around how an intervention is designed and implemented can imply this. I think this needs explanation and clarification. Moreover, this demonstrates that simply asking is not enough to modify behaviour or to expect adherence.

Reply 12: This is an excellent point. We have revised the text to make it clear that the instructions about the intervention asked participants to focus on inner obstacles: “Second, although the instructions asked participants to focus on overcoming inner obstacles….” We also note that simply asking is not enough to change behaviour “This is shows that simply asking is not always enough to modify behaviour”.

Round 2

Reviewer 2 Report

Comments and Suggestions for Authors

Accept in the current format.

Reviewer 3 Report

Comments and Suggestions for Authors

Thank you to the authors for considering reviewer feedback and making adjustments based on this feedback. The quality of has improved relevant to both the structure and communication of the dialogue within the manuscript.